# Valorization of Agricultural Waste as a Chemiresistor H_2_S-Gas Sensor: A Composite of Biodegradable-Electroactive Polyurethane-Urea and Activated-Carbon Composite Derived from Coconut-Shell Waste

**DOI:** 10.3390/polym15030685

**Published:** 2023-01-29

**Authors:** Aamna Bibi, Karen S. Santiago, Jui-Ming Yeh, Hsui-Hui Chen

**Affiliations:** 1Department of Chemistry, Center for Nanotechnology and R & D Center for Membrane Technology at Chung Yuan Christian University, Chung Li 32023, China; 2Department of Chemistry, College of Science, Research Center for the Natural and Applied Sciences, University of Santo Tomas, Manila 1015, Philippines; 3Department of Molecular Science and Engineering, National Taipei University of Technology, Taipei 10608, China

**Keywords:** biodegradable-electroactive, PUU, H_2_S gas sensor, activated carbon, composite, room temperature

## Abstract

In this study, a high-performance H_2_S sensor that operates at RT was successfully fabricated using biodegradable electroactive polymer-polyurethane-urea (PUU) and PUU-activated-carbon (AC) composites as sensitive material. The PUU was synthesized through the copolymerization of biodegradable polycaprolactone diol and an electroactive amine-capped aniline trimer. AC, with a large surface area of 1620 m^2^/g and a pore diameter of 2 nm, was derived from coconut-shell waste. The composites, labeled PUU-AC1 and PUU-AC3, were prepared using a physical mixing method. The H_2_S-gas-sensing performance of PUU-AC0, PUU-AC1, and PUU-AC3 was evaluated. It was found that the PUU sensor demonstrated good H_2_S-sensing performance, with a sensitivity of 0.1269 ppm^−1^ H_2_S. The H_2_S-gas-sensing results indicated that the PUU-AC composites showed a higher response, compared with PUU-AC0. The enhanced H_2_S-response of the PUU-AC composites was speculated to be due to the high surface-area and abounding reaction-sites, which accelerated gas diffusion and adsorption and electron transfer. When detecting trace levels of H_2_S gas at 20 ppm, the sensitivity of the sensors based on PUU-AC1 and PUU-AC3 increased significantly. An observed 1.66 and 2.42 times’ enhancement, respectively, in the sensors’ sensitivity was evident, compared with PUU-AC0 alone. Moreover, the as-prepared sensors exhibited significantly high selectivity toward H_2_S, with minimal to almost negligible responses toward other gases, such as SO_2_, NO_2_, NH_3_, CO, and CO_2_.

## 1. Introduction

H_2_S is a hazardous gas, even at the sub-ppm level, and may lead to poisoning or deaths at high concentrations [1]. People may be exposed to this gas through geothermal activities, organic decomposition, treatment plants, and as a by-product or intermediate of various industries. H_2_S is one of most dangerous gases in the workplace according to the Occupational Safety and Health Administration [2,3]. Hence, monitoring of this type of contaminant in the environment is of utmost significance [4].

Intrinsic conducting polymers (ICPs) exhibit the electronic properties of metals and semiconductors; they appeared as promising gas-sensing materials in the early 1980s [5]. Given its environmental stability, facile synthesis, unique redox behavior, low toxicity, and simple acid/base doping properties, polyaniline (PANI) is one of the most useful ICPs [6,7]. PANI has various applications, including as a corrosion inhibitor [8], supercapacitor [9], electrochemical sensor, [10] and gas sensor [11], on the basis of its reversible redox behavior and dope/de-dope properties. 

With its easy processing from solution into uniform thin films, PANI has elicited considerable interests in gas-sensing applications [12,13,14,15,16,17]. Thin PANI films change their conductivity upon exposure to various polar and nonpolar gases [18]. Conductivity depends on the ability to transport and hop charge carriers along the polymer backbone and between polymer chains. Thus, any interactions with PANI that alter either of these processes will affect conductivity [19]. PANI is a biocompatible polymer with concentration (≥400 mg L^−1^)-dependent toxicity [20]. PANI reinforced with various carbon fillers has been used to improve gas-sensing performance [21,22,23,24,25,26]. However, considering the poor solubility of PANI, aniline oligomer-based polymers have attracted intensive research interests in the past decade [27]. In 2005, Zhao et al. designed a floating-gate ion-sensitive field-effect transistor that used doped aniline trimer as a sensing material; this transistor could sense up to 70 ppm of NH_3_ gas [28]. In 2009, aniline trimer-based polyimide as a H_2_S-gas sensor was presented by Santiago and Yeh et al. [29]. In 2020, Yeh et al. reported the effect of the surface morphology of electroactive polyamic acid with aniline trimer in the main chain on H_2_S-gas-sensing performance [27]. 

Recent years have witnessed an unparalleled rate of technological advancement and a consequent increase in people’s reliance on electronics and devices. The extremely fast production of electronic and smart sensing-devices has begun to represent a significant additional source of pollution for the environment in the form of electronic waste or e-waste, and to exert a drastic impact on ecosystems [18,30,31,32,33]. Compared with the global production and consumption of plastics, the electronic market makes a minimal contribution to e-waste; nevertheless, its contribution should not be disregarded [34]. Biodegradable electronic devices [33,34,35,36,37,38] that exhibit environmental safety and disposability may be an effective solution to the e-waste issue by simultaneously decreasing the cost of the recycling operation and the health risk associated with harmful emissions [38]. Biopolyesters, such as polycaprolactone (PCL), polyhydroxyalkanotes, and polylactic acid, which are obtained from renewable resources, comprise a large class of eco-friendly/green polymers. These polymers have commercial applications in optoelectronic devices, dielectric materials, electromagnetic interference shielding, and the synthesis of sensors [39,40,41]. 

Thus, biodegradable environmental sensors [42,43] are anticipated to achieve a green environment. Dong and Li et al. fabricated a biodegradable micromotor based on PCL that exhibited fluorescence gas-sensing properties toward HCl and NH_3_ gas [44]. In addition, some researchers have developed gas sensors by utilizing polymer blends based on biodegradable polymers and conducting polymers; these sensors have been successfully employed in gas detection. For example, Low et al. prepared a chemosensor by using PCL as the insulating polymer host in a PANI-nanofiber/PCL-blend system for sensing water vapor, NH_3_, and NO_2_ gases [45]. Similarly, Macagnano et al. designed an eco-friendly conductive chemosensor for NH_3_-gas detection. This goal was achieved by employing a biodegradable electrospun-nanofibrous scaffold based on a polymer blend that contained PANI and poly(3-hydroxybutyrate) (PANI/PHB) [18]. Moreover, biodegradable electroactive copolymers (BEPs) have been reported by various groups for use as a bioactive scaffold for tissue regeneration, bone regeneration, and drug delivery [46,47,48,49].

To the best of the authors’ knowledge, no such report has been made on BEP application in gas detection. The objective of the current study is to create and investigate an eco-friendly conductive sensor for H_2_S gas based on BEP and its composites. To achieve this goal, BEP was synthesized through the copolymerization of PCL diol (a biopolyester) and an electroactive segment of aniline trimer, i.e., amine-capped aniline trimer (ACAT), with methylene diphenyl diisocyanate (MPDI) as a chain extender. Various biodegradable sensors based on PCL [50,51,52,53,54,55,56] have been reported in the literature. BEP composites were prepared by reinforcing BEP with activated carbon (AC) derived from coconut-shell waste. AC composites derived from various sources have been reported in gas detection [57,58,59,60,61]. AC derived from coconut shell is amorphous in nature; it is virtually harder, dust-free, with high purity and density, and resistant to attrition. Moreover, the production of AC is financially viable, due to the abundant supply of coconut shell [62]. To the best of our knowledge, BEP-AC had never been tested in a H_2_S-gas-sensing application until the present study. Thus, a series of sensors is fabricated and tested.

## 2. Experimental Section

### 2.1. Chemicals and Instrumentation

Aniline (99%, ACROS, Lisbon, Portugal) was distilled, prior to use. Ammonium persulfate (APS, 97.0%, J. T. baker, Avantor performance materials, LLC, Randor, PA, USA), 3,6-dimethyl-1,4-dioxane-2,5-dione (NMP) (Macron fine chemicals, Avantor performance materials, LLC, Randor, PA, USA), hydrochloric acid (HCl, 37%) and ammonium hydroxide (NH_4_OH, 25%, Honeywell Fluka, Wunsrofer strasse 40, Seetze, Germany), 4,4′-diaminodiphenylamine sulfate hydrate, Dibutyltin dilaurate (DBTDL) and Methylene diphenyl diiscocyante (MPDI (TCI, Tokyo, Japan), Polycaprolactone-diol (PCL, Mn = 2000, Sigma Aldrich, St. Louis, MO, USA), PBS (Dulbecco’s phosphate buffer saline modified, without calcium chloride, magnesium chloride, Sigma Aldrich, Saint Louis, MO, USA) were used as received without further purification. All the gases, including H_2_S (50 ppm +N_2_), SO_2_ (50 ppm + N_2_), NH_3_ (50 ppm + N_2_), NO_2_ (50 ppm + N_2_), CO (50 ppm + N_2_), CO_2_, N_2_ and air (Chian Hong Gas Co., LTD, Hong Kong, China) used in this study were used as received. Indium-tin oxide (ITO, Ruilong Optoelectronics, Miaoli Taiwan) was used to create an inter-digitated electrode device (thickness 0.375 mm, 12 pairs of electrodes, 0.3 mm spacing and l × w of 20 × 20 mm) for the study of gas sensing.

The Bruker 300 spectrometer, FTIR spectrometer (JASCO, FT/IR-4200,Tokyo, Japan) and Bruker Daltonics IT mass spectrometer model Esquire 2000 (Leipzig, Germany) were used to record 1H-NMR, FTIR and mass spectra. The degradability test was performed in a DENG YNG water bath G-10 (Dogger, Taipei, Taiwan). The oven (PFY400, Dengyng instruments Co., LTD, Kaohsiung, Taiwan) was used for AC preparation. GPC-150CV (Waters International Corporation Taipei, Taiwan) and a JASCO V-750 (Sunway scientific corporation, Taipei Taiwan) were used for molecular-weight determination and UV-visible absorption spectroscopy. Cyclic voltammetry (CV) and contact angle were determined using the AutoLab PGSTAT 204 (Ω Metrohm Autolab, KM Utrecht, the Netherlands) electrochemical work station and a First Ten Angstroms FTA 125, respectively. The morphology and nature of the composite was confirmed using scanning electron microscopy (SEM, JEOL JSM-7600F, Tokyo, Japan). The N_2_ adsorption–desorption isotherm (BET, Micromeritics ASAP-2010, GA USA) was used to determine the surface are and pore diameter of AC. Conductivity of the sensors was recorded on a four-point probe (LRS4-TK1, KeithLink Technology, Taipei, Taiwan). A laser engraver (Flux, Beambox, Taipei, Taiwan) was used for ITO-electrode fabrication. H2S sensing of all ITOs coated with PUU-AC0 and PUU-AC composites were also determined, using the sensing device constructed in our lab. The sensor’s response was measured on a Keithley 2450 SourceMeter (Keithlink Technology Co., Ltd., Taipei, Taiwan).

### 2.2. Synthesis of N, N’-bis(4′-aminophenyl)-1,4-Quinonenediimine (ACAT)

ACAT was synthesized as shown in Appendix A [63].

### 2.3. Synthesis of Polyurethane Urea (PUU-AC0)

PUU was synthesized via a two-step process [64], as shown in Figure 1. PCL diol (3 mmol) was melted at 100 °C in a three-neck flask under N_2_ atmosphere for 2 h. PMDI (1.5 equivalent) was then added to the flask, and stirred to obtain an NCO-terminated polyurethane pre-polymer. After 30 min, 3 mmol of ACT was dissolved in NMP, and the reaction continued to be refluxed further for 48 h. Thereafter, PUU films were prepared using sequential heating-steps in an oven.

### 2.4. Preparation of Activated Carbon (AC) from Coconut-Shell Waste

Figure 1 (top) showed the preparation of AC. The natural biomass coconut-shell was cleaned three times with distilled water, followed by overnight drying in an oven at 60 °C. Firstly, it was cut into small pieces and heated at 300 °C in an oven under N_2_ for 2 h, to obtain pre-carbonized biochar. Later, this was followed by washing (methanol, ethanol, distilled water) and drying. For the activation step, pre-carbonized biochar was mixed with ZnCl_2_ at a 1:10 weight ratio, and carbonized at 800 °C for 2 h under N_2_ flow in a tube furnace. The temperature ramp was set to 1.5 °C min^−1^ during carbonization. The as-synthesized activated carbon (AC) was then washed with H_2_O: Ispropanol (1:1) and vacuum dried at 80 °C for 6 h [65].

### 2.5. Preparation of PUU-AC Composites

A simple physical mixing-method was used to prepare the PUU-AC composites. First, 1 wt% solution of PUU was prepared in NMP via magnetic-stirring at room temperature. Subsequently, 1 wt% and 3 wt% AC was added to the previous solution, followed by overnight stirring. The as-prepared composites were labeled PUU-AC1 and PUU-AC3, respectively.

### 2.6. Sensor Preparation

An interdigitated ITO electrode was used as laser engraver to carve grooves into the ITO glass (12 pairs of electrodes, 0.3 mm spacing, and l × w of 20 × 20 mm) creating a nonconductive open-circuit on both sides of the glass. The spin-coating technique was used in the construction of ITO coated with PUU-AC0, PUU-AC1, and PUU-AC3. First, 1 wt% solution of the given samples PUU-AC0, PUU-AC1, and PUU-AC3 was prepared in NMP via stirring at room temperature, as shown in Figure 1 (bottom). Thereafter, thin films (0.005 mm) were prepared by spin-coating 200 µL of the respective solution on ITO, at a spin rate of 1600 rpm, followed by overnight drying at room temperature. The advantages of spin coating are to produce very fine, thin, and uniform coating. Using the spin-coating method, the desired thickness of the film can be achieved. On the other hand, the drop-casting method is used to prepare a film which is not uniform and also not thin. In addition, there will be the possibility of cracks after evaporation of the solvent. The dip-coating method involves the immersion of the substrate in the respective solution, which cannot be employed in the gas-sensing experiment.

### 2.7. H_2_S Sensing Experiment

In the gas-sensing experiment, the films spin-coated onto ITO were exposed to H_2_S-gas concentration ranging from 1 ppm to 50 ppm [12] at room temperature. The ITO sensor fixed in a gas chamber was linked to the electrometer (Keithley 2450 SourceMeter, Keithlink Technology Co., Ltd.) to measure the response, as shown in Figure 2. All the measurements were performed at room temperature (25 ± 0.5 °C) after a steady state was achieved. The carrier gas (air) was blended with H_2_S gas from the cylinder at a fixed concentration (50 ppm in N_2_) to determine the dependency of ITO responses on gas concentration. The gas chamber had a total-gas-flow value set to 1000 sccm. Each measurement was taken by flushing H_2_S gas through the measurement chamber for an interval of 150 s, followed by cleaning the sensor in which H_2_S gas was replaced with air, until the baseline was achieved. ITO responses were calculated as normalized resistance (R_a_−R_g_), where R_a_ and R_g_ denote resistance under air and a given analyte, respectively.

## 3. Results and Discussion

Characterization: ACAT was synthesized and fully characterized via FTIR. The ion-trap mass spectrum is shown in Appendix A.

The nitrogen adsorption–desorption isotherms of AC are presented in Figure 3a. The profile of the isotherm curve is Type I, which is characteristic of a microporous material, i.e., pores with a diameter of 2 nm (inset in Figure 3a). The surface area of AC was determined to be 1620 m^2^/g, with a pore volume of 0.91 cm^3^/g. The micropore area was 1392 m^2^/g, while single-point adsorption total-pore-volume was 1.05 cm^3^/g. Figure 3b shows the Raman spectrum of AC. A typical Raman spectrum of AC is characterized by two bands, as shown in Figure 3b. In the case of AC, the G-band observed at 1580 cm^− 1^ is related to the stretching vibration of sp^2^ carbon in a hexagonal lattice, while the D-band that appeared at 1332 cm^−1^ is associated with the disordered or amorphous carbon atoms.

FTIR was used for the structural determination of PUU-AC0 and its composites. Figure 4a shows the FTIR spectra of PCL, PMDI, ACAT, and PUU-AC0. The absence of an absorption band at 2260 cm^−1^ of the –NCO group indicated that all of the –NCO groups were incorporated into the PUU-AC0 copolymer. The single absorption peak at 3422 cm^−1^ in PUU-AC0 demonstrated the formation of a urea group. Most of the absorption bands found in PUU-AC0 agreed with PCL diol except at 1539 cm^−1^ and 1500 cm^−1^, exhibiting the presence of quinoid and benzenoid ring from ACAT. The carbonyl-group peak stretching from the urethane bond and urea linkage appeared at 1673 cm^−1^ and 1602 cm^−1^, respectively. The molecular weight of PUU-AC0 was determined via gel-permeation-chromatography analysis with a value of Mw = 252,437, Mn = 160,550, and PDI = 1.5.

The FTIR spectra of PUU-AC0, PUU-AC1, and PUU-AC3 are shown in Figure 4b. In general, biomass-derived AC has surface oxygenated-functional groups. The surface functional groups present on AC from coconut shell were determined via FTIR, as shown in Figure 4b. AC presented prominent absorption bands at 3196, 2984, 2637, 1651, 1585, 1219, 970 and 828 cm^−1^, as shown in Figure 4b. The intramolecular bonded O–H group of alcohol appeared at 3196 cm^−1^. The peaks at 2984 cm^−1^ and 2637 cm^−1^ indicated the C–H and O–H stretching vibrations of the carboxylic-acid functional group, respectively. The absorption peaks at 1651 cm^−1^ and 1585 cm^−1^ were assigned to the C = C bond of the aromatic ring. The absorption peak at 1219 cm^−1^ was attributed to C–O in carboxylic acids, alcohols, phenols, and esters, or to the P = O bond in phosphate esters.

In addition, the bending vibration at 828 cm^−1^ was ascribed to the C–H bond from a highly substituted aromatic ring. Overall, the AC may contain the functional groups of C–O, O–H, –CH_2_– or –CH_3_, C = C, and highly substituted aromatic rings [66,67,68,69]. The composites, i.e., PUU-AC1 and PUU-AC3 presented all the characteristic peaks of PUU-AC0 and AC, indicating the covering of the PUU network on the surface of AC. However, a slight shift occurred in the characteristic bands, due to the incorporation of AC into the polymer matrix. The interactions between PUU-AC0 and AC may be ascribed to the slight shift in the observed bands.

Figure 5 shows the scanning-electron-microscopy (SEM) images of AC, PUU-AC0, PUU-AC1, and PUU-AC3. SEM imaging of the AC derived from coconut shell (Figure 5a) revealed that the production of rich porous structure was favored by the high ratio of ZnCl_2_/sample [70]. The SEM image shows cavities, pores, and more rough surfaces on AC. The holes and cave-type opening on the AC surface increased the surface available for adsorption of the given analyte. Figure 5b shows the SEM image of PUU with no evident surface morphology. As shown in Figure 5c,d, the morphology of PUU-AC0 changed after the addition of AC. During the blending process, PUU-AC0 diffused into the micropores of AC, as indicated by the absence of agglomeration.

## 4. Biodegradability

The quantitative data of the weight loss [71] for PUU-AC0 films in PBS was given in Figure 6a. As can be seen, over the period of six months, the film show progressive mass-loss, with a degradation rate of 3.33 ± 0.0002%.

## 5. Electrical Properties

Table 1 indicates that the electrical conductivity of the undoped and doped as-synthesized sensors was determined from four-point probe measurements. HCl is commonly used for the doping process. In the current study, H_2_S gas was also used for doping (because it is used as a dopant in gas-sensing properties). Thus, two types of dopants, 1 M HCl and 20 ppm of H_2_S_(g)_ (50 ppm + N_2_), were used to dope thick and thin films, respectively. An electrical conductivity of 6.95 × 10^−7^, 8.46 × 10^−7^, and 9.94 × 10^−7^ S/cm was exhibited by PUU-AC0, PUU-AC1, and PUU-AC3, respectively. After doping with 1 M HCl for 10 min, the electrical conductivity of PUU-AC0 increased up to 1.60 × 10^−6^ S/cm. An increase of 3.5 and 3.7 times was observed for the PUU-AC1 and PUU-AC3 thick films, respectively. Moreover, the electrical conductivity determined for the thin films of PUU-AC0, PUU-AC1, and PUU-AC3 was 1.22 × 10^−5^, 7.92 × 10^−5^, and 3.6 × 10^−4^ S/cm, respectively. After doping with 20 ppm of H_2_S gas for 10 min, a tremendous increase of ×5, ×9, and ×10.5 was observed for PUU-AC0, PUU-AC1, and PUU-AC3, respectively, as indicated in Table 1.

CV was used to determine the effect of AC on the electrical properties of PUU-AC composites (Figure 6b). Prior to the evaluation of PUU-AC0, PUU-AC1, and PUU-AC3 as H_2_S-gas sensor, current–voltage (I–V) curves (Figure 6c) were obtained in the absence of H_2_S.

CV measurements were made at a scan rate of 10 mV s^−1^ from 0–0.8 V, which presented two distinct peaks at 0.44–0.56 V and 0.22–0.30 V, indicating the electron transfer of AC-doped PUU. The area under the CV peaks increased with an increase in the AC proportion in the composites. The I–V characteristics of the PUU-AC0, PUU-AC1, and PUU-AC3 electrodes are presented in Figure 6c. The I–V curves for these films are linear, indicating an ohmic contact between the electrode and PUU-AC films. The σ of the films increased from PUU to PUU-AC3. The CV and I–V characteristics of the PUU and PUU-AC composite electrodes indicated that the AC-doped PUU resulted in an improved charge-transfer, due to the synergistic effect between the PUU matrix and AC, which, in turn, improved the sensor performance for H_2_S detection.

## 6. H_2_S-Gas-Sensing Properties

The H_2_S-sensing properties of all the sensors were studied at room temperature. The sensors exhibited an apparent difference in response with and without the filler, i.e., AC.

## 7. Response and Sensitivity

The dynamic response and recovery-curves toward H_2_S with different concentrations (from 50 ppm to 1 ppm) of the PUU-AC0, PUU-AC1, and PUU-AC3 sensors are shown in Figure 7a. After the injection of H_2_S gas, resistance decreased sharply, returning to the original baseline as H_2_S gas was replaced with air [72]. As the concentration of H_2_S gas decreased, the response-value decreased correspondingly. For the PUU-AC0 sensor, the response value was 3 for detecting 20 ppm H_2_S gas. For the PUU-AC1 and PUU-AC3 sensors, the response values were as high as 4 and 7.27, respectively. For the 20 ppm H_2_S gas, the response value was 1.33, 2.42 times higher than that of PUU-AC0. The PUU-AC0, PUU-AC1, and PUU-AC3 sensors exhibited an evident response value of 0.11, 0.52, and 1, respectively, at a lower detection limit of 1 ppm. The large surface area of AC may be ascribed to the difference in sensor response. Considering the availability of a large number of active sites in the sensing layer, the relative response increased. Hence, the performance of the sensors increased in the following order: PUU-AC0 < PUU-AC1 < PUU-AC3.

Figure 7b shows the response magnitudes of the PUU-AC0, PUU-AC1, and PUU-AC3 film sensors versus different H_2_S concentrations. The linear fitting equations of y = 0.1269x + 0.3249, y = 0.2047x + 0.4712, and 0.2706x + 1.8642 were determined for PUU-AC0, PUU-AC1, and PUU-AC3 respectively. The values of 0.9901, 0.9919, and 0.9847 were observed for correlation coefficients of the fitted data (*R*^2^), respectively. Moreover, Figure 7c shows the bar plot for the sensor’s sensitivity (S, [ppm^−1^]), which was calculated as the slope of the normalized sensor response, R_a_−R_g_. Evidently, compared with the PUU-AC0 sensor, the PUU-AC1 and PUU-AC3 sensors displayed better sensitivity. A sensitivity of 0.1269 ppm^−1^ was measured for PUU-AC0, which was increased up to 0.2047 ppm^−1^ for the PUU-AC1 sensor. The PUU-AC3 sensor exhibited the highest sensitivity, of 0.2706 ppm^−1^, which was 2.1 times and 1.32 times higher than those of PUU-AC0 and PUU-AC1, respectively. The water contact-angle measurements of PUU-AC0, PUU-AC1, and PUU-AC3 were 79.7°, 73.3°, and 72.2°, respectively, indicating their hydrophilic nature. The hydrophilic groups may have developed intermolecular interactions with polar H_2_S gas, thus increasing the response of the sensor [27,73].

## 8. Response/Recovery Time

For gas-sensing properties, an important parameter is response–recovery time. Upon exposure to gas, adsorption and desorption occur simultaneously. Thus, the adsorption/desorption rate is solely responsible for the response–recovery time. Figure 8 shows the response–recovery time of the as-prepared sensors as a function of the H_2_S gas at 20 ppm. Figure 8 shows that at 20 ppm, the quick response-time for PUU-AC0, PUU-AC1, and PUU-AC3 sensor was 55, 34, and 32 s (±0.05), respectively. The response-time window for PUU-AC0 toward H_2_S gas ranging from 1 ppm to 50 ppm was 120–15 s, as shown in Appendix A. However, this window was reduced to 57–16 s and 55–15 s for the PUU-AC1 and PUU-AC3 sensors, respectively. The short response-time window for the composites may be attributed to the higher absorption rate, due to a larger surface area compared with that of PUU-AC0. At 20 ppm, however, ITO coated with PUU-AC3 and PUU-AC1 required a longer recovery-time, of 315 s and 345 s (±0.05), respectively, compared with PUU-AC0, which had a recovery time of 200 s. The large surface area allowed the adsorption of a sufficient amount of H_2_S gas molecules to dope, probably leading to a higher adsorption-rate [74]. However, de-doping the sensing material took a longer time, i.e., the desorption rate was considerably slower, which may lead to the poor recovery time (Appendix A) of the composites.

Based on the literature search, the performance of the H_2_S-gas sensor based on different conductor materials or composite materials is listed in Table 2. The PUU-AC3 synthesized in this work showed a good response for H_2_S at a room temperature, with a good sensitivity of 0.2724 ppm^−^ of H_2_S gas.

## 9. Selectivity

Another salient feature for gas-sensing execution is selectivity. This feature was studied upon exposure to various gases, including H_2_S, SO_2_, NO_2_, NH_3_, CO, and CO_2_ at 20 ppm, at room temperature. Figure 9a illustrates the fact that all the sensors exhibited a similar trend, by showing the highest sensing selectivity toward H_2_S gas. Moreover, the PUU-AC3 sensor exhibited the highest response to H_2_S, the value of which was approximately 4.5 times higher than that for the other test gases.

## 10. Effect of Humidity

Figure 9b shows the effect of humidity on the response of the as-prepared sensors to H_2_S gas at 20 ppm. The relative humidity (RH) range was 60% to 100%. The response value (of 3, 4.03, and 7.27) at 40% RH was set as the standard value (100%) for PUU-AC0, PUU-AC1, and PUU-AC3, respectively. As shown in the figure, moisture negatively affects the performance of the sensors, resulting in an attenuation of more than 95% of the initial value at 100% RH.

## 11. Stability

Stability (Figure 9c) was determined by the continuous measurements of the response values of the PUU-AC0, PUU-AC1, and PUU-AC3 sensors to 20 ppm of H_2_S gas at room temperature, for 30 days. The response value of all the sensors decreased with the passage of time, which coincided with the biodegradable nature of the as-prepared sensors.

## 12. Repeatability

Repeatability and stability are crucial parameters for the practical application of gas sensors, as shown in Figure 10. Figure 10 shows the response curves of PUU-AC0, PUU-AC1, and PUU-AC3 at 20 ppm of H_2_S gas, at room temperature. All the sensors demonstrated repeatability, without any attenuation in response.

## 13. Gas-Sensing Mechanism

Doping/de-doping plays a crucial role in the gas-sensing mechanism of ICP-based [79] sensors. Similar to PANI, aniline oligomer-based PUU may also be doped via redox reaction or protonation. H_2_S is a reducing gas. It may undergo ionization in the presence of water (Equation (1)), as shown in Figure 11. The resulting proton ions subsequently dope the PUU polymer reversibly [80], as shown in Equation (2).
(1)$$H_{2}S\overset{\overset{H_{2}O}{\to }}{\leftarrow }{\mathrm{HS}}^{-}+H^{+}$$
(2)$$\mathrm{PUU}+H^{+}\overset{\overset{\;\;\;\;\;\;\;\;\;\;}{\to }}{\leftarrow }{\mathrm{PUUH}}^{+}$$
(3)$${\mathrm{PUUH}}^{+}\overset{\overset{\mathrm{air}}{\to }}{\leftarrow }\mathrm{PUU}$$

The equilibrium shifted toward right and left under and after H_2_S-gas exposure, respectively. In addition to the doping effect mentioned above, the functional groups present in PUU may also exhibit weak intermolecular interactions with H_2_S gas [27].

## 14. Conclusions

The H_2_S-gas-sensing performance of the biodegradable electroactive polymer (PUU-AC0) and composites (PUU-AC1 and PUU-AC3) was investigated at room temperature. A sensor was fabricated using the PUU-AC0, PUU-AC1, and PUU-AC3. This sensor was able to detect H_2_S with a concentration ranging from 1 ppm to 50 ppm. All the given sensors demonstrated good selectivity, rapid response, and good reproducibility against H_2_S gas. For 20 ppm H_2_S, the PUU-AC0, PUU-AC1, and PUU-AC3 sensors exhibited a high gas-sensing response of 3, 5, and 7.3, respectively, with a response-time window of 55–32 s. Among all of these, the PUU-AC3 sensor exhibited the smallest response-time, while PUU-AC0 showed the smallest recovery-time. Given the synergistic effect between PUU and AC, the PUU-AC1 and PUU-AC3 gas-sensors demonstrated a highly sensitive and selective gas-sensing performance toward H_2_S gas relative to PUU-AC0, with a humidity of 60%. All sensors showed stability of up to 30 days, and thus may be employed for practical application. Our experimental results indicate that porous PUU-AC is a potential candidate for high-performance H_2_S-sensing materials.

## Data Availability

Not applicable.

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
