# Peer review of "Valorization of Agricultural Waste as a Chemiresistor H2S-Gas Sensor: A Composite of Biodegradable-Electroactive Polyurethane-Urea and Activated-Carbon Composite Derived from Coconut-Shell Waste"

_polymers, 2023, doi:10.3390/polym15030685_

Round 1

Reviewer 1 Report

Reviewer’s Comments:

The manuscript “H2S gas sensor based on biodegradable-electroactive polyurethane-urea and activated carbon composite derived from coconut shell waste” is a very interesting work. In this work, a high-performance H2S sensor was successfully prepared using biodegradable electroactive polymer polyurethane urea (PUU) and PUU-activated carbon (AC) composites as sensitive material at room temperature. The PUU was synthesized through the copolymerization of biodegradable polycaprolactone diol and an electroactive amine-capped aniline trimer. AC with a large surface area of 1620 m2 /g and a pore diameter of 2 nm was derived from coconut shell waste. The composites, labeled PUU-AC1 and PUU-AC3, were prepared using a physical mixing method. The H2S gas-sensing performance of PUU-AC0, PUU-AC1, and PUU-AC3 was evaluated. The PUU sensor demonstrated good H2S-sensing performance with a sensitivity of 0.1269 ppm−1 . The H2S gas-sensing results indicated that the PUU-AC composites showed higher response compared with PUU-AC0. The enhanced H2S response of the PUU-AC composites was speculated to be due to the high surface area and abounding reaction sites, which accelerated gas diffusion and adsorption and electron transfer. The results are consistent with the data and figures presented in the manuscript. While I believe this topic is of great interest to our readers, I think it needs major revision before it is ready for publication. So, I recommend this manuscript for publication with major revisions.

1. In this manuscript, the authors did not explain the importance of the activated carbon composite in the introduction part. The authors should explain the importance of activated carbon composite.

2) Title: The title of the manuscript is not impressive. It should be modified or rewritten it.

3) Correct the following statement “When detecting 20 ppm of H2S gas, the sensitivity of PUU-AC1 and PUU-AC3 increased by 1.66 and 2.42 times, respectively, compared with PUU-AC0 alone. The as-prepared sensors exhibited significantly high selectivity toward H2S but only minor responses toward other gases, such as SO2, NO2, NH3, CO, and CO2”.

4) Keywords: The activated carbon composite is missing in the keywords. So, modify the keywords.

5) Introduction part is not impressive. The references cited are very old. So, Improve it with some latest literature like 10.1016/j.jallcom.2021.162012, 10.1039/D0RA09390D

6) The authors should explain the following statement with recent references, “After the injection of H2S gas, resistance decreased sharply, returning to the original baseline as H2S gas was replaced by air”.

7) Add space between magnitude and unit. For example, in synthesis “21.96g” should be 21.96 g. Make the corrections throughout the manuscript regarding values and units.

8) The author should provide reason about this statement “The large surface area allowed the adsorption of a sufficient amount of H2S gas molecules to dope, probably leading to a higher adsorption rate”.

9. Comparison of the present results with other similar findings in the literature should be discussed in more detail. This is necessary in order to place this work together with other work in the field and to give more credibility to the present results.

10) Conclusion part is very long. Make it brief and improve by adding the results of your studies.

11) There are many grammatic mistakes. Improve the English grammar of the manuscript.

Author Response

Reviewer 1:

  1. In this manuscript, the authors did not explain the importance of the activated carbon composite in the introduction part. The authors should explain the importance of activated carbon composite.

The application of activated-carbon composites mentioned in the introduction part last paragraph.

2) Title: The title of the manuscript is not impressive. It should be modified or rewritten it.

                   The manuscript title was revised.

3) Correct the following statement “When detecting 20 ppm of H2S gas, the sensitivity of PUU-AC1 and PUU-AC3 increased by 1.66 and 2.42 times, respectively, compared with PUU-AC0 alone. The as-prepared sensors exhibited significantly high selectivity toward H2S but only minor responses toward other gases, such as SO2, NO2, NH3, CO, and CO2”.

                   The sentence was revised.

4) Keywords: The activated carbon composite is missing in the keywords. So, modify the keywords.

                   The activated carbon composite was present in key words.

5) Introduction part is not impressive. The references cited are very old. So, Improve it with some latest literature like 10.1016/j.jallcom.2021.162012, 10.1039/D0RA09390D

                   The above mentioned papers were added in the introduction part.

6) The authors should explain the following statement with recent references, “After the injection of H2S gas, resistance decreased sharply, returning to the original baseline as H2S gas was replaced by air”.

                   The recent reference was added for the above mentioned statement.

7) Add space between magnitude and unit. For example, in synthesis “21.96g” should be 21.96 g. Make the corrections throughout the manuscript regarding values and units.

                   The space was added in the manuscript where needed.    

8) The author should provide reason about this statement “The large surface area allowed the adsorption of a sufficient amount of H2S gas molecules to dope, probably leading to a higher adsorption rate”.

                   The reference was added for the statement.

  1. Comparison of the present results with other similar findings in the literature should be discussed in more detail. This is necessary in order to place this work together with other work in the field and to give more credibility to the present results.

The table was added comparing the results from literature with our results.

10) Conclusion part is very long. Make it brief and improve by adding the results of your studies.

                   Conclusion part was revised.

11) There are many grammatic mistakes. Improve the English grammar of the manuscript.

                   The English editing service was availed for grammatical corrections.

Reviewer 2 Report

The manuscript titled " H2S gas sensor based on biodegradable-electroactive polyurethane-urea and activated carbon composite derived from coconut shell waste" is meaningful and can be useful for a better understanding of H2S gas sensor. I recommend accepting the manuscript after all the following corrections are incorporated by the author:

1.      In Title: “biodegradable” term is used; however the author has not provided sufficient reason for including this term

Does the author have performed any biodegradability test for prepared sensing material.

2.      Please improve the introduction section.

Kindly cite the following in the appropriate place.

https://www.pubs.thesciencein.org/journal/index.php/jmns/article/view/282

3.      In line 168, the author has mentioned the use of spin coating technique to deposit 0.005mm of sensing material over substrate using 1wt% solution of sample in NMP, however (a) how much time (hours) the solution is stirred? (b) and at what spin rate this thickness is achieved? moreover, please specify various reason of using this spin coating technique over other technique of deposition such as drop coating, dip coating etc.

4.      Please specify how hydrophilic nature of sensing material effects the sensitivity (as mentioned in line 337 and figure 7 c).

5.      The author is advised to characterized the prepared sensing material using other characterization technique as well such as XRD, EDS etc.

6.      Please elaborate the reason of having a relatively good recovery of the sensor resistance to the initial value for PUU-AC3 sensor.

7.      Please explain the technique used for preparation of IDEs over ITO substrate.  

8.      I request the author to kindly elaborate the sensing mechanisms.

Kindly read and cite the recent article published on the detection of H2S gas sensor.

https://iopscience.iop.org/article/10.1088/1361-6439/ac82f8/meta

Author Response

Reviewer 2:

  1. In Title: “biodegradable” term is used; however, the author has not provided sufficient reason for including this term. Does the author have performed any biodegradability test for prepared sensing material?

The biodegradability test was performed and mentioned in the manuscript as figure xx.

  1. Please improve the introduction section. Kindly cite the following in the appropriate place. https://www.pubs.thesciencein.org/journal/index.php/jmns/article/view/282

The above mentioned reference was added in the introduction section

  1. In line 168, the author has mentioned the use of spin coating technique to deposit 0.005mm of sensing material over substrate using 1wt% solution of sample in NMP, however (a) how much time (hours) the solution is stirred? (b) and at what spin rate this thickness is achieved? moreover, please specify various reason of using this spin coating technique over other technique of deposition such as drop coating, dip coating etc.

                   The detail was added in section 2.6.

  1. Please specify how hydrophilic nature of sensing material effects the sensitivity (as mentioned in line 337 and figure 7 c).

The reference was added for the statement mentioned in the comment.

  1. The author is advised to characterized the prepared sensing material using other characterization technique as well such as XRD, EDS etc.

It will take time (atleast one month) to do these characterization techniques. As I need to prepare material first.

  1. Please elaborate the reason of having a relatively good recovery of the sensor resistance to the initial value for PUU-AC3 sensor.

After experimental results, I also tried to find a reason behind this, but unfortunately could not found this.

  1. Please explain the technique used for preparation of IDEs over ITO substrate.  

                   A detailed procedure was added in section 2.6

  1. I request the author to kindly elaborate the sensing mechanisms. Kindly read and cite the recent article published on the detection of H2S gas sensor. https://iopscience.iop.org/article/10.1088/1361-6439/ac82f8/meta.

The equations were added to elaborate the gas sensing mechanism further. And the above mentioned reference was added in the introduction section. 

Reviewer 3 Report

The authors describe a degradable PUU material with a specific response to hydrogen sulfide. This paper describes the synthesis process of the functional material in detail and discusses the effect of the addition of active carbon on the response performance. The authors also introduced the humidity effect, stability, response mechanism of the material, etc. 

General comments:

1.       The novel material introduced by the authors is biodegradable. It is a promising research topic for environmental reasons. In the stability section, it can see that all three PUU-ACs materials have lost their response to H2S. The degradation conditions of this experiment are different from the general degradation conditions https://doi.org/10.3390/ijms10093722. Is it can be provable that the degradation of PUU occurs in the absence of microorganisms?

2.       In Figure 5, there is a labeling error on line 254, which should be 'c)'. In the part of nitrogen adsorption–desorption, the results of the specific surface area have fully demonstrated that the AC synthesized by the authors has a very good porous structure. Microscale SEM characterization (fig. 5a) lacks statistical significance compared to specific surface results. Moreover, the 10-micron-scale porosity in the picture is not the same as the concept of porosity (2 nm) that the author wants to express. In the absence of EDS mapping or image annotation, it is difficult for readers to figure out what these surface morphologies represent. Are these small objects in Figure 5 (c,d,e) AC particles or polymer particles? The lack of detailed SEM pictures does not serve to support the author's argument here. In fact, it can be placed in the supporting information.

3.       The authors of this article seem to define response time as the time it takes to reach a certain signal strength. Due to the higher sensitivity of PUU-AC3, the time required to reach a certain signal strength will obviously be shorter. However, this definition challenges common sense. Generally speaking, the response time is defined as the time required to reach from 5% to 95% of the maximum signal intensity at a certain concentration. In this way, the result of response speed needs to be readjusted. Clearly, the response of PUU-AC3 should be the slowest. This also echoes the results in Figure 10. In Figure 10, the time for PUU-AC0 to complete the 5 tests is less than 2000 seconds, while both PUU-AC1 and PUU-AC3 exceed 2500 seconds.

In addition, the author did not give PUU-AC1 and PUU-AC3 enough time to reach maximum intensity in figure 8. we can see that AC1 and AC3 had not reached the maximum signal intensity (plateau) before stopping the supply of H2S. It is also puzzling that AC1 and AC3 still have signals rising after stopping H2S. Can this be understood as H2S was cut off before AC1 and AC3 reached the maximum response, while the residual H2S in the dead volume of the system maintains the continuation of the response leading to a signal increase?

Minor comments:

1.       Page 14 line 381, “40% RH” should be 60% RH

2.       Please check the abbreviation for Methylene di phenyl diiscocyante. MPDI and PMDI have been used in this paper, including in FTIR notation. 

3. Please check the sorting labels capitalization in Figure 5 and Figure 6

Author Response

Reviewer 3:

  1. The novel material introduced by the authors is biodegradable. It is a promising research topic for environmental reasons. In the stability section, it can see that all three PUU-ACs materials have lost their response to H2S. The degradation conditions of this experiment are different from the general degradation conditions https://doi.org/10.3390/ijms10093722. Is it can be provable that the degradation of PUU occurs in the absence of microorganisms?

The biodegradability test for PUU performed and mentioned in the manuscript in figure xx by a procedure mentioned in the following article. http://dx.doi.org/10.1016/j.msec.2014.07.061

  1. In Figure 5, there is a labeling error on line 254, which should be 'c)'. In the part of nitrogen adsorption–desorption, the results of the specific surface area have fully demonstrated that the AC synthesized by the authors has a very good porous structure. Microscale SEM characterization (fig. 5a) lacks statistical significance compared to specific surface results. Moreover, the 10-micron-scale porosity in the picture is not the same as the concept of porosity (2 nm) that the author wants to express. In the absence of EDS mapping or image annotation, it is difficult for readers to figure out what these surface morphologies represent. Are these small objects in Figure 5 (c,d,e) AC particles or polymer particles? The lack of detailed SEM pictures does not serve to support the author's argument here. In fact, it can be placed in the supporting information.

                    The SEM image of PUU-AC0 has no surface morphology as shown in figure 5b. While AC is a porous structure as shown in figure 5a. After AC addition to PUU-AC0, PUU-AC0 diffused in to pores of AC and no clear morphology observed.

  1. The authors of this article seem to define response time as the time it takes to reach a certain signal strength. Due to the higher sensitivity of PUU-AC3, the time required to reach a certain signal strength will obviously be shorter. However, this definition challenges common sense. Generally speaking, the response time is defined as the time required to reach from 5% to 95% of the maximum signal intensity at a certain concentration. In this way, the result of response speed needs to be readjusted. Clearly, the response of PUU-AC3 should be the slowest. This also echoes the results in Figure 10. In Figure 10, the time for PUU-AC0 to complete the 5 tests is less than 2000 seconds, while both PUU-AC1 and PUU-AC3 exceed 2500 seconds.

In addition, the author did not give PUU-AC1 and PUU-AC3 enough time to reach maximum intensity in figure 8. we can see that AC1 and AC3 had not reached the maximum signal intensity (plateau) before stopping the supply of H2S. It is also puzzling that AC1 and AC3 still have signals rising after stopping H2S. Can this be understood as H2S was cut off before AC1 and AC3 reached the maximum response, while the residual H2S in the dead volume of the system maintains the continuation of the response leading to a signal increase?

The quick response time is defined as the time required by the sensor to show a response to given analyte. PUU-AC3 showed a quick response with smallest time window as compared to PUU-AC0 and PUU-1.

 Figure 10 shows the repeatability having 5 consecutive responses.  Response shown in figure showed the response time along with recovery time. Recovery time for PUU-AC composites is longer as compared to PUU-AC0. AC1 as shown in figure 8.

H2S gas doped the given sensors leading to a decrease in the resistance. When the H2S gas cut off, de-doping takes place. De-doping takes longer time for composites with larger surface area as compared to polymer itself. That’s why composites still showed a response even after replacement of H2S gas by air and also showed the longer recovery time.

Round 2

Reviewer 3 Report

My comments were answered